# Mutation status and prognostic value of KRAS and NRAS mutations in Moroccan colon cancer patients: A first report

Fatima El agy [1,2]*, Sanae el Bardai[2], Ihsane El Otmani[1,2,3], Zineb Benbrahim[4], Ibn Majdoub Hassani Karim[5], Khalid Mazaz[5], El Bachir Benjelloun[5], Abdelmalek Ousadden[5], Mohammed El Abkari[6], Sidi Adil Ibrahimi[5], Laila Chbani[1,2]

1 Faculty of Medicine and Pharmacy, Laboratory of Biomedical and Translational Research, Sidi Mohamed Ben Abdellah University, Fez, Morocco, 2 Laboratory of Anatomic Pathology and Molecular Pathology, University Hospital Hassan II, Sidi Mohamed Ben Abdellah University, Fez, Morocco, 3 Unit of Medical Genetics and Oncogenetics, University Hospital Hassan II, Sidi Mohamed Ben Abdellah University, Fez, Morocco, 4 Department of Oncology, University Hospital Hassan II, Sidi Mohamed Ben Abdellah University, Fez, Morocco, 5 Department of General Surgery, University Hospital Hassan II, Sidi Mohamed Ben Abdellah University, Fez, Morocco, 6 Department of Gastroenterology, University Hospital Hassan II, Sidi Mohamed Ben Abdellah University, Fez, Morocco

* Fatima.elagy@usmba.ac.ma

**Data Availability Statement:** All relevant data are within the paper.

**Funding:** The authors received no specific funding for this work.

## Abstract

This study aimed to estimate the incidence of KRAS, NRAS, and BRAF mutations in the Moroccan population, and investigate the associations of KRAS and NRAS gene mutations with clinicopathological characteristics and their prognosis value. To achieve these objectives, we reviewed medical and pathology reports for 210 patients. RAS testing was investigated by Sanger sequencing and Pyrosequencing technology. BRAF (exon 15) status was analyzed by the Sanger method. The expression of MMR proteins was evaluated by Immunohistochemistry. KRAS and NRAS mutations were found in 36.7% and 2.9% of 210 patients, respectively. KRAS exon 2 mutations were identified in 76.5% of the cases. RAS-mutated colon cancers were significantly associated with female gender, presence of vascular invasion, classical adenocarcinoma, moderately differentiated tumors, advanced TNM stage III-IV, left colon site, higher incidence of distant metastases at the time of diagnostic, microsatellite stable phenotype, lower number of total lymph nodes, and higher means of positive lymph nodes and lymph node ratio. KRAS exon 2-mutated colon cancers, compared with KRAS wild-type colon cancers were associated with the same clinicopathological features of RAS-mutated colon cancers. NRAS-mutated patients were associated with lower total lymph node rate and the presence of positive lymph node. Rare RAS-mutated tumors, compared with wild-type tumors were more frequently moderately differentiated and associated with lower lymph node rate. We found that KRAS codon 13-mutated, tumors compared to codon 12-mutated tumors were significantly correlated with a higher death cases number, a lower rate of positive lymph, lower follow-up time, and poor overall survival. Our findings show that KRAS and NRAS mutations have distinct clinicopathological features. KRAS codon 13-mutated status was the worst predictor of prognosis at all stages in our population.

**Competing interests:** The authors have declared that no competing interests exist.

## Introduction

Ras-family G-proteins transduce growth factor signals, such as EGF receptor (EGFR) signaling pathway [1]. KRAS and NRAS mutations appear early in colorectal carcinogenesis (aberrant crypt foci lacking dysplasia and polypsis) leading to constitutive signaling and downstream activation of mitogen-activated protein kinase (MAPK) and phosphoinositide 3- kinase (PI3K) dependent pathways [2–4], and contribute also to tumor progression in association with others genetic alterations [5]. Numerous studies have reported that KRAS and NRAS mutations occur respectively in 45% and 5–8% of worldwide colorectal cancer patients [6]. The overwhelming majority of KRAS and NRAS mutations occur in codons 12 and 13 (90%) of KRAS gene and less frequently in codons 59, 61, 117 and 146 of KRAS and NRAS genes. In the field of metastatic colorectal treatment, previous studies have demonstrated that KRAS exon 2 mutations have a negative effect on response to anti epidermal growth factor receptor (EGFR) monoclonal antibodies (cetuximab and panitumumab) [7]. In fact, it have been confirmed that patients with KRAS codon 12 and 13 mutants tumors do not respond to anti-EGFR monoclonal antibodies. In contrast, patients with WT KRAS exon 2 tumors significantly benefited from the treatment. Recently, mutations in KRAS outside exon 2 (KRAS exons 3,4) and in other downstream effectors of the EGFR signalling pathway, such as NRAS have been reported to be also associated with resistance to anti-EGFR monoclonal antibodies [8]. For this reason, since 2013 full RAS WT state was recommended to be confirmed in all mCRC patients before initiating treatment with Anti- EGFR monoclonal antibodies, panitumumab and cetuximab. Previous reports have demonstrated that KRAS exon 2 mutations might be associated with many clinicopathological features. Christophe and all [9] found in a cohort of 41514 patients that KRAS mutations appear frequently in older patients with a predominance of male gender and located in right (coecum). Mucinous differentiation has been reported to be correlated with Kras mutations by some studies, whereas others reported no relationship [9,10]. KRAS, NRAS and BRAF mutations are mutually exclusive; however KRAS-mutant CRCs also have PIK3CA and APC mutations [11]. Furthermore KRAS exon 3 mutations are associated with deficient mismatch repair status and Lynch syndrome. Several local studies had analyzed data on KRAS- mutant CRCs. Some studies have demonstrated that KRAS gene mutated colorectal cancers were correlated with worse overall survival [12]. Although, others found noassociation [13].

Up until now, the association between all RAS status (exon 2, 3 and 4 of KRAS and NRAS genes) and tumor features has been investigated by few studies [14]. Recently, the Molecular Cancer Genetics Platform of Poitiers (French) has published interesting results about the clinicopathological features of every KRAS and NRAS mutation in a cohort of 1735 French colorectal cancer patients, enrolled from 28 hospital molecular genetics platforms throughout France [15]. However, this type of study has been never established in Moroccan colon cancer patients. Our study aimed to estimated the incidence of KRAS and NRAS mutations (KRAS exons 2/3/4 and NRAS exons 2/3/4) in Moroccan population, to identify the clinicopathological characteristics of KRAS and NRAS gene mutations and to evaluate their prognosis value in colon cancer.

## Materials and methods

### Ethics

This study protocol was reviewed and approved by Hassan II University Hospital Ethics Committee of FEZ, Morocco, under reference no. 13/18. All patients gave written informed consent before the start of the study.

## Patients

A total of 210 patients were enrolled in this study at Hassan II University Hospital, Fez, Morocco, from 2015 to 2020. Medical charts were prospectively and retrospectively reviewed, and patients were recruited using the following selection criterion: (a) patients with histologically confirmed primary adenocarcinoma (b) all cases with pathologic I-IV stage colon cancer, (c) patients with prognostic information and who underwent surgical resection for CC tumor at the surgery department of Hassan II University Hospital. Patients were excluded if their records were incomplete, without histological confirmation of colon adenocarcinoma and if they had rectal cancer. Demographic and clinicopathological information (e.g. age, gender, tumor grade, tumor site histological subtype, disease stage, and numbers of examined regional lymph nodes. . .) and follow up data, were obtained from the patient's medical records and pathology reports. Overall survival was defined as the time from the start of diagnosis until death or until the last follow-up.

Molecular testing was performed at the molecular pathology department of Hassan II University Hospital. Before 2016, only KRAS exon 2 mutations have been analyzed. Since 2016, we have started testing complete RAS status systematically, for every metastasis colon cancer patient when considered for anti-EGFR therapy.

## DNA extraction

Tumoral DNA was extracted from paraffin- embedded tumor sections. The blocks with higher proportion of tumors cells was selected by a pathologist on hematoxylin, safran and eosin- stained slides. From the selected FFPE tumor block 4–8 sections of 5 μm thickness were obtained for DNA extraction using the QIAamp DNA FFPE Tissue Kit (Invitrogen), and according to the manufacturer's instructions. DNA concentration (ng/ul) was assessed by Qubit fluorometer.

## Molecular analysis

KRAS (exon 2, 3 and 4) and NRAS (exon 2, 3, and 4) mutations were analyzed using polymerase chain reaction (PCR) amplification and pyrosequençing on the Qiagen PyroMark Q24 device according to the CE-IVD-marked therascreen RAS Pyro Kit Handbook.

**Direct sequencing.** PCR was performed in 25 ul volume containing 10 ng of template DNA, 12× PCR mix platinium, 12.5 pmol primers, and 50 umol Mgcl2. Primer sequences used for PCR are presented in Table 1. PCR products were purified using ExoSAP® kit according to the manufacturer's protocols. The purified PCR products were sequenced using the direct sequencing with BigDye Terminator V3.1 Cycle Sequencing Kit (ABI Prism) and analyzed on

**Table 1. Primer sequences used for PCR.**

| Primer name | Primer sequence |
|---|---|
| KRAS-ex 2- F | 5'-GGTGGAGTATTTGATAGTGTA- 3' |
| KRAS-ex 2- R | 5'-TGCATATTACTGGTGCAGACC- 3' |
| KRAS-ex 3- F | 5'-AGTAAAAGGTGCACTGTAATAA-3' |
| KRAS-ex 3- R | 5'-ATAATAAGCTGACATTAAGGAG-3' |
| KRAS-ex 4- F | 5'-TGTTACTAATGACTGTGCTATAACTTTT-3' |
| KRAS-ex 4- R | 5'-TATGCTATACTATACTAGGAAATAAAA-3' |
| NRAS-ex2-F | 5'-ATGACTGAGTACAAACTGGTGGTGGTTGGAGCA-3' |
| NRAS-ex2-R | 5'-CACTTTGTAGATGAATATGATCCCACCATAGAG-3' |
| NRAS-ex3-F | 5'-GATTCTTACAGAAAACAAGTGGTTA-3' |
| NRAS-ex3-R | 5'-CATTTGCGGATATTAACCTCTACAG-3' |
| NRAS-ex4-F | 5'-GGAGCAGATTAAGCGAG-3' |
| NRAS-ex4-R | 5'-TCAGCCAAGACCAGACAG-3' |

Applied Biosystems 3500Dx Genetic Analyzer (Applied Biosystem). Results were visualized using Sequencing Analysis software v5.4.

**Pyrosequencing.** The TheraScreen® KRAS Pyro kit (for KRAS codons 12 and 13), and the TheraScreen® RAS Extension Pyro kit (for KRAS codons 59/61, 117 and 146, and NRAS codons 12, 13, 59, 61, 117 and 146) (Qiagen, GERMANY), were used for RAS mutations testing, according to the manufacturer's instructions. Briefly, 5 µl of template DNA (2–10 ng of genomic DNA) were amplified by polymerase chain reaction (PCR) in a 20 µl volume containing 12.5 µl of PyroMark® PCR Master Mix 2x, 2.5 µl of Coral Load Concentrate 10x, 4 µl of nuclease-free water, and 1 µl of the corresponding set of PCR primers (Qiagen). 10 µl of PCR products were immobilized to Streptavidin Sepharose High Performance beads (Qiagen) to prepare the single-stranded DNA. The corresponding sequencing primers were allowed to anneal to the DNA using a PyroMark Q24 plate and a vacuum workstation (Qiagen). The sequences were analyzed using Pyromark Q24 software in the AQ analysis mode. In each run, a negative control (without template DNA) and an unmethylated control DNA, provided by the kit as a positive control for PCR and sequencing reactions were included.

## BRAF molecular testing

BRAF testing was performed for 200 patients using Sanger sequencing as described previously (KRAS and NRAS mutations testing). Briefly, DNA was amplified by PCR (Master Mix (2X) kits) according to the manufacturer's protocols and using the following forward primer (5'-AC GAACGAGACTATCCTTTTAC-3') and reverse primer (5'- CATTGAGTCGTCGTAGAGTCCC-3'). PCR products were purified using ExoSAP® kit according to the manufacturer's protocols. The purified PCR products were sequenced using the direct sequencing with BigDye Terminator V3.1 Cycle Sequencing Kit (ABI Prism) and analyzed on Applied Biosystems 3500Dx Genetic Analyzer (Applied Biosystems).

## Detection of MMR protein expression

The mismatch repair tumor status (MSS or MSI) was established by immunohistochemistry (IHC) to detect the intact or the loss expression of the MMR proteins (MLH1, PMS2, MSH2, and MSH6). The IHC study was assessed on unstained formalin-fixed paraffin-embedded (FFPE) tumor tissue sections of 4 $\mu$m thickness, on the automated immunostainer Ventana Benchmark ULTRA. We have employed monoclonal antibodies specific for each MMR protein, MLH1 (G168-728/CELL MARQUE), MSH2 (G219-1129/CELL MARQUE), MSH6 (44/CELL MARQUE), and PMS2 (MRQ-28/CELL MARQUE).Adjacent normal tissue (lymphocytes or normal glandular cells) was used as an internal control for positive staining.

## Statistical analysis

Clinical, pathological, and molecular variables collected at baseline were described as means and standard deviation (sd's) for quantitative variables and percentages for qualitative variables. Associations between mutational status and tumor characteristics were assessed using the $\chi2$-test or Fisher's exact test for categorical variables and using unpaired t-test for continuous variables. Tests were statistically significant when $p<0.05$.

PFS and OS according to KRAS status were analyzed using the Kaplan-Meier method to estimate the probability of survival and survival difference with the use of the log-rank test. All reported P-values were the result of two-sided tests, with $P<0.05$ considered statistically significant. Multivariate analysis was performed using a Cox proportional hazard model to identify prognostic factors. Factors that were significant and nearly significant ($P<0.1$) were included in univariate analysis. Statistical analysis was performed using the IBM SPSS Statistic 21.

## Results

### Molecular and clinicopathological features of patients

A total of 210 patients with locally and advanced CC, were enrolled in this study. Patients and tumor characteristics are summarized in Table 2. 97 (46.2%) were Women and 113 (53.8%) were Men with a mean age of 55.56 years (range, 16–92 years). The left-sided colon cancers were the most frequent tumors diagnosed in our cohort (n = 143, 68.1%), compared to right-sided CC (n = 67, 31.9%), indicating a predominance of left CC in our population.

The most common histological subtype of CC was adenocarcinoma with 172 cases (81.9%) and less frequent subtype was mucinous adenocarcinoma with 26 (12.4%) cases. Others sub-types were diagnosed only in 12 cases (5.7%). 99 (47.1%) tumors out of 210 were moderately differentiated, 92 (43.8%) and 19 (9.0%) were well and poorly differentiated, respectively. A vascular invasion was seen in 37 (17.6%) tumors. Perineural invasion was identified in 22 (10.7%) tumors. In this study, the mean number of removed Lymph Nodes was 18.40 (rang, 1–53), and 153 (76.9%) patients had more than 12 dissected LN. Positive LNs were identified in 83 (39.5%) patients (mean = 1.74; rang, 1–18). Regarding the pathologic stage, 126 (60%) of tumors were classified as stage I-II and 84 (40%) as stage III-IV. 83 (39.5%) of patients study were classified as metastatic group.

The mean follow-up time of overall survival was 46.74 months (rang, 3–132 months). From 210 patients' colon cancer, 53 (25.4%) were died. Moreover, the rate of recurrence was 75 35.7% (n = 75). Regarding molecular characteristics of our cohort, the MSI status was reported in 17 patients (14.5%) and KRAS and NRAS mutations in 83 patients (39.5%). Although, from 200 patients, we did not find any BRAF mutation.

### Mutations frequencies and current alterations

KRAS and NRAS mutation were detected in 39.5% (83/210) of tumor cases examined, of which 36.7% (77/210) and 2.9% (6/210) were occurred in KRAS and NRAS gene respectively. BRAF mutations analysis was performed on 200 tumors, while we did not find any case with BRAF mutation. Interestingly, simultaneous mutations were not found in this study Table 3.

All data according to the site of mutations are presented in Table 4. In the KRAS-mutant CC group, 88.3% (n = 68/77) of mutations were in exon 2, 1.3% in exon 3 (1/77), and 10.4% (8/77) in exon4. Among mutations in KRAS exon 2, 76.5% (52/68) of cases had single mutations in codon 12 and 23.5% (16/68) of cases in codon 13. The most frequent mutant type in exon 2 of KRAS gene was G12D (19/77, 24.7%), followed by G12V (17/77, 22.1%) and G13D (13/77, 16.9%), others mutation were also found in our study (G12C, G12A, G12R, G13V, G13R) but were less frequent, Table 4. Exon 3 KRAS mutations were found in one case. The mutation was located in codon 61 (Q61L) and it accounted for 1.3% (1/77) of KRAS-mutant CC group. In exon 4, 5 mutations (5/77, 6.5%) were located in codon 146 and 3 mutations in codon 117 (3/77, 3.9%). The main mutant types were K117N and A146T with a percentage of (3/77, 3.9%) followed by A146P and A146V (1/77, 1.3%).

NRAS mutations occurred in 33.3% (2/6) of cases in exon 2, and in 66.7% (4/6) of cases in exon 3. The main mutant types were G12D in codon 12 of exon 2 (2/6, 33.3%), Q61K in codon 61of exon 3 (2/6, 33.3%) and Q61L also in the same site (2/6, 33.3%), Table 4.

### Association between KRAS and NRAS mutations and clinicopathological features

Table 5 demonstrates the relationship between KRAS and NRAS mutations status and clinicopathological parameters. Compared with RAS wild-type tumors, RAS mutant tumors

**Table 2. Clinicopathological and molecular characteristics of patients.**

| Characteristics | Total |
|---|---|
| **Age:** | |
| Mean (±SD) | 55.56 (±14.3) |
| ≤55 | 92 (43.8%) |
| ≥55 | 118 (56.2%) |
| **Gender** | |
| Female | 97 (46.2%) |
| Male | 113 (53.8%) |
| **Tumor site** | |
| Right colon | 67 (31.9%) |
| Left colon | 143 (68.1%) |
| **Histologic subtype:** | |
| Adenocarcinoma | 172 (81.9%) |
| Mucinous adenocarcinoma | 26 (12.4%) |
| Others | 12 (5.7%) |
| **Histologic grade:** | |
| Well | 92 (43.8%) |
| moderate | 99 (47.1%) |
| Poor | 19 (9.0%) |
| **Vascular invasion:** | |
| Yes | 37 (17.6%) |
| No | 173(82.4%) |
| **perineural invasion:** | |
| Yes | 22 (10.7%) |
| No | 184 (89.3%) |
| Distant metastases (M) | |
| M0 | 127 (60.5%) |
| M1 | 83 (39.5%) |
| Number of metastases: | |
| 1 | 51 (68.0%) |
| ≥2 | 24 (32.0%) |
| Number of removed lymph nodes: | |
| Mean (± SD) | 18.40 (± 9.67) |
| <12 | 46 (23.1%) |
| ≥12 | 153 (76.9%) |
| Average | 1–53 |
| Positive lymph node: | |
| Mean (± SD) | 1.74 (± 3.63) |
| Presence | 83 (39.5%) |
| Absence | 127 (60.5%) |
| Average | 1–18 |
| Lymph node ratio: | |
| mean (± SD) | 0.16 (±0.38) |
| range | 0.00–3.25 |
| Disease stages | |
| I-II | 126 (60%) |
| III-IV | 84 (40%) |
| MSI status | |

(*Continued*)

**Table 2.** (Continued)

| Characteristics | Total |
|---|---|
| MSS | 100 (85.5%) |
| MSI | 17 (14.5%) |
| BRAF mutation: | |
| Presence | 0 (0%) |
| Absence | 200(100%) |
| RAS mutation | |
| Presence | 83(39.5%) |
| Absence | 127(60.5%) |
| **follow-up time (months):** | |
| Mean (SD) | 46.74 (±34.53) |
| Range | 3–132 |
| **Recurrence** | |
| (+) | 75 (35.7%) |
| (−) | 135 (64.3%) |
| **Mortality** | |
| Death cases | 53 (25.4%) |
| Censored cases | 156 (74.6%) |

were statistically associated with female gender (P = 0.003), presence of vascular invasion (P = 0.03), classical adenocarcinoma (P = 0.01), moderately differentiated tumors (0.04) and advanced TNM stage III-IV (P = 0.02). KRAS and NRAS mutations tended also to be located in the left colon (P = 0.009), to have a higher incidence of distant metastases at the time of diagnostic (P = 0.03) and occurred more frequently in tumors with microsatellite stable phenotype (P = 0.02). Regarding nodal counts, RAS mutant tumors subgroup was significantly associated with lower number of total lymph nodes examined, than the Wild-type subgroup (P <0.001). 61.1% of the mutated RAS patients have more than 12 lymph nodes examined, compared with 85.8% in the wild-type RAS patients (P <0.001). Although, the means of positive LNs and Lymph node Ratio (LNR) were significantly higher in RAS mutant tumors (P = 0.03); (P<0.001). Interestingly when analyzing our cohort, the number of recurrence and death cases was significantly higher in RAS-mutant tumors subgroup. In contrast, there were no significant associations with other clinical and pathological features (age, perineural invasion).

**Table 3. Mutational status of patients with CC by genes.**

| Gene alterations | Number (%) |
|---|---|
| **KRAS** (n = 210): | |
| Wildtype (n = 133) | 66.3% |
| Mutated (n = 77) | 36.7% |
| **NRAS** (n = 210): | |
| Wildtype (n = 204) | 97.1% |
| Mutated (n = 6) | 2.9% |
| **BRAF** (n = 200): | |
| Wildtype | 100% |
| Mutated | 0% |

**Table 4. Frequency and distribution of KRAS, NRAS, and BRAF mutations.**

| Gene | Exon | Nucleotide substitution | Codon substitution | Amino acid substitution | Number | % |
|------|------|------------------------|-------------------|------------------------|--------|---|
| | | c.35G>A | GGT>GAT | p.G12D | 19 | 24.7% |
| | | c.35G>T | GGT>GTT | p.G12V | 17 | 22.1% |
| | 2 | c.34G>T | GGT>TGT | p.G12C | 9 | 11.7% |
| | | c.35G>C | GGT>GCT | p.G12A | 5 | 6.5% |
| | | c.34G>C | GGT>CGT | p.G12R | 2 | 2.6% |
| | | c.38 G>A | GGC>GAC | p.G13D | 13 | 16.9% |
| KRAS | | c.38G>T | GGC>GTC | p.G13V | 2 | 2.6% |
| | | c.37G>C | GGC>CGC | p.G13R | 1 | 1.3% |
| | 3 | c.182A>T | CAA>CTA | p.Q61L | 1 | 1.3% |
| | 4 | c.351A>T | AAA>AAT | p.K117N | 3 | 3.9% |
| | | c.436G>C | GCA>ACA | p.A146T | 3 | 3.9% |
| | | c.436G>A | GCA>CCA | p.A146P | 1 | 1.3% |
| | | c.437C>T | GCA>GTA | p.A146V | 1 | 1.3% |
| NRAS | 2 | c.35G>A | GGT>GAT | p.G12D | 2 | 33.3% |
| | 3 | c.181C>A | CAA>AAA | p.Q61K | 2 | 33.3% |
| | | c.182A>T | CAA>CTA | p.Q61L | 2 | 33.3% |
| | 4 | any | any | any | 0 | 0% |
| BRAF | 15 | WT | | pV600E | 0 | 0% |

## Association between KRAS and NRAS mutation subtypes and clinicopathological features

Secondly, we investigated the association between KRAS and NRAS mutation subtypes and clinicopathological parameters. As shown in Table 6, female gender, left localization, classical adenocarcinoma, vascular invasion, presence of positive lymph node and advanced disease stage were found to be associated with KRAS mutated colon cancers as compared with KRAS wild-type colon cancers. More KRAS mutated colon cancers had a higher incidence of metastatic disease at diagnosis (P = 0.04). At all stages, the mean of examined lymph nodes was significantly higher in KRAS wild-type tumors than KRAS mutated tumors (P<0.001). In addition, 62.1% of the mutated KRAS tumors have more than 12 lymph nodes examined, compared with 84.2% in the wild-type RAS patients (P <0.001). Furthermore the presence of positive lymph node was significantly associated with KRAS mutated tumors (P = 0.003). Moreover, KRAS mutations were more likely to appear in tumors with microsatellite-stable phenotype (P = 0.03). Generally, the incidence of recurrence and death cases was significantly higher in KRAS-mutated colon cancers. Compared to NRAS wild- type patients, NRAS-mutated patients, were more likely to exhibit lower total lymph node rate (P = 0.04). This subgroup of tumors was also marked by the presence of positive lymph node (83.3%, P = 0.01). Although, there were no significant associations in others clinicopathological characteristics between NRAS mutant and WT patients.

We then looked at the associations between rare mutations (exons 3 and 4 in KRAS gene and exons 2, 3 and 4 of NRAS gene) and patients' clinicopathological characteristics (Table 4). Rare RAS-mutated tumors compared with WT tumors, were more frequently moderately differentiated (86, 7% versus 44, 1%, P = 0.003) and was associated with lower total lymph node rate (P = 0.007). In addition, 53.8% of these tumors have more than 12 lymph nodes removed, compared with 21.5% in RAS wild-type tumors (P = 0.04). Furthermore, Rare KRAS and NRAS mutations were detected more frequent in left colon with a difference close to significance (P = 0.06).

**Table 5. Clinicopathological characteristics according to KRAS and NRAS mutations status in 210 colon cancer patient.**

| Characteristics | RAS Wild-type | RAS Mutants | p-value |
|---|---|---|---|
| **Age:** | | | |
| Mean (±SD) | 55.3 (±14.3) | 55.9 (±15.7) | 0.7 |
| ≤55 | 57 (44.9%) | 35 (42.2%) | 0.1 |
| ≥55 | 70 (55.1%) | 48 (57.8%) | |
| **Gender** | | | |
| Female | 49 (38.6%) | 48 (57.8%) | **0.003** |
| Male | 78 (61.4%) | 35 (42.2%) | |
| **Tumor site** | | | |
| Right colon | 48(37.8%) | 19 (22.9%) | **0.009** |
| Left colon | 79 (62.2%) | 64 (77.1%) | |
| **Histologic subtype:** | | | |
| Adenocarcinoma | 107 (84.3%) | 65 (78.3%) | **0.01** |
| Mucinous adenocarcinoma | 10 (7.9%) | 16 (19.3%) | |
| Others | 10(7.9%) | 2 (2.4%) | |
| **Histologic grade:** | | | |
| Well | 58 (45.7%) | 34 (41.0%) | **0.04** |
| moderate | 54 (42.5%) | 45 (54.2%) | |
| Poor | 15 (11.8%) | 4 (4.8%) | |
| **Vascular invasion:** | | | |
| Yes | 15 (11.8%) | 22 (26.5%) | **0.006** |
| No | 112 (88.2%) | 61(73.5%) | |
| **perineural invasion:** | | | |
| Yes | 14 (11.0%) | 8 (10.1%) | 0.1 |
| No | 113 (89.0%) | 71(89.9%) | |
| Distant metastases (M) | | | |
| M0 | 82 (64.6%) | 45 (54.2%) | **0.03** |
| M1 | 45 (35.4%) | 38 (45.8%) | |
| Number of metastases: | | | |
| 1 | 31 (73.8%) | 20 (60.6%) | **0.04** |
| ≥2 | 11 (26.2%) | 13 (39.4%) | |
| Number of removed lymph nodes, mean ± SD | 20.43 ±9.37 | 14.83 ±9.19 | <**0.001** |
| <12 | 18 (14.2%) | 28 (38.9%) | <**0.001** |
| ≥12 | 109 (85.8%) | 44 (61.1%) | |
| Positive lymph node: | | | |
| Mean (± SD) | 1.34 (±3.70) | 2.44 (±3.4)1 | **0.03** |
| Presence | 29 (22.8%) | 35 (48.6%) | |
| Absence | 98 (77.2%) | 37 (51.4%) | <**0.001** |
| Lymph node ratio, mean ± SD | 0.07 ±0.18 | 0.32 ±0.55 | <**0.001** |
| Disease stages | | | |
| I-II | 57 (44.9%) | 27 (32.5%) | **0.02** |
| III-IV | 70 (55.1%) | 56 (67.5%) | |
| MSI status | | | |
| MSI | 15 (19.5%) | 2 (5.0%) | **0.02** |
| MSS | 62 (80.5%) | 38 (95.0%) | |
| BRAF mutation: | - | - | - |
| **follow-up time (months):** | | | |

*(Continued)*

**Table 5.** (Continued)

| Characteristics | RAS Wild-type | RAS Mutants | p-value |
|---|---|---|---|
| **Mean** | 50.08 | 41.69 | **0.04** |
| SD | ±34.19 | ±34.63 | |
| **Recurrence** | | | |
| (+) | 36 (28.3%) | 39 (47.0%) | **0.003** |
| (−) | 91 (71.7%) | 44 (53.0%) | |
| **Mortality** | | | |
| Death cases | 26(20.5%) | 27 (32.9%) | **0.01** |
| Censored cases | 101 (79.5%) | 55 (67.1%) | |

## Association between KRAS exon 2 mutation subtypes and clinicopathological features

A summary of the main clinicopathological features of KRAS codon 12 mutants and KRAS codon 13 mutants compared to RAS wild-Type colon cancer is shown in Table 7. In the majority of cases KRAS codon 12 mutated tumors were associated with female gender (P = 0.01), classical adenocarcinoma (P = 0.02), advanced TNM stage (P = 0.04) and presence of positive lymph node (P = 0.004). KRAS codon 12 mutated colon cancers were also associated with a significantly lower total LN count (P <0.001), and 63.6% of patients in this subgroup have more than 12 lymph nodes examined, compared with 80.6% in the KRAS codon12-wild- type patients (P = 0.01). While, lymph node metastasis (P = 0.02) and LNR (P <0:001) rate was significantly higher in KRAS codon 12 mutated tumors. Interestingly, KRAS codon 12 mutations were correlated with a higher recurrence rate (P = 0.02). Tumors with KRAS codon 13 mutants, in comparison with RAS wild-type colon cancers were more likely to exhibit lower total LN rate (P = 0.05), and the majority of LN were significantly less than 12 (P = 0.05). Moreover, KRAS codon 13 mutants tumors were correlated with a higher risk of death (P = 0.003) Table 6.

When comparing the different KRAS exon 2 mutation (codon 12 versus codon 13) groups to each other, we found that KRAS codon 13-mutated tumors were significantly correlated with a higher death cases number (P = 0.03) and lower follow-up time (p = 0.03), lower rate of positive lymph node and lower rate of LNR. Although, it did not reach the statistical significance, KRAS codon 13 mutated tumors tended to be located in the left colon (P = 0.09) and to have more than 1 metastasis site (P = 0.08) compared to the same features in KRAS codon 12 mutated tumors.

## Prognostic significance of genetic alterations

Follow up data from all 210 CC patients were included in the survival analysis. The median length of the follow-up period was 34.00 months (range 3–132 months), and there were 53 CC-related deaths. The Kaplan–Meier analysis revealed that RAS mutant patients exhibited the shortest OS compared with RAS wild-type patients (78.18 vs. 104.29, P = 0.006). When assessing the prognostic value of RAS genes (KRAS and NRAS), we found that KRAS mutant patients had a strong association with poorer OS compared to RAS-WT patients (76.81 vs 104.14, P = 0.003). Although, NRAS mutations, and rare mutations did not show any association with the patients' OS (P = 0.6, and P = 0.8).

We also evaluated OS by KRAS exon 2 mutation subtypes, and we compared survival among three groups: codon12/13 WT CC, codon 12 mutant CC, and codon 13 mutant CC. Among these three groups, OS was significantly poorer for codon 13 mutant CC patients than

**Table 6. Association between KRAS and NRAS mutation subtypes and clinicopathological parameters.**

| Characteristics | KRAS mutant | KRAS wild type | p | NRAS Mutant | NRAS wild type | p | Rare mutations | wild-type status | p |
|---|---|---|---|---|---|---|---|---|---|
| **Age** | | | | | | | | | |
| Mean ((±SD) | 55.8 ± 16.0 | 55.3 ±14.3 | 0.7 | 56.5 ±13.0 | 55.5 ±14.9 | 0.8 | 50.2 ±16.0 | 55.8 ±14.7 | 0.1 |
| ≤55 | 33 (42.9%) | 59 (44.4%) | 0.1 | 2 (33.3%) | 90 (44.1%) | 0.4 | 7 (46.7%) | 85 (43.6%) | 0.5 |
| >55 | 44 (57.1% | 74 (55.6%) | | 4 (66.7%) | 114 (55.9%) | | 8 (53.3%) | 110 (56.4%) | |
| **Gender** | | | | | | | | | |
| Female | 45(58.4%) | 52 (39.1%) | **0.003** | 3 (50.0%) | 94 (46.1%) | 0.4 | 7 (46.7%) | 90 (46.2%) | 0.5 |
| Male | 32(41.6%) | 81(60.9%) | | 3 (50.0%) | 110 (53.9%) | | 8 (53.3%) | 105 (53.8%) | |
| **Tumor site** | | | | | | | | | |
| Right colon | 18 (23.4%) | 49(36.8%) | **0.01** | 1 (16.7%) | 66 (32.4%) | 0.2 | 2 (13.3%) | 65 (33.3%) | 0.06 |
| Left colon | 59 (76.6%) | 84 (63.2%) | | 5 (83.3%) | 138 (67.6%) | | 13 (86.7%) | 130 (66.7%) | |
| **Histologic subtype** | | | | | | | | | |
| Adenocarcinoma | 60 (77.9%) | 112(84.2%) | **0.02** | 5 (83.3%) | 167(81.9%) | 0.5 | 12 (80.0%) | 160 (82.1%) | 0.4 |
| Mucinous adenocarcinoma | 15(19.5%) | 11 (8.3%) | | 1(16.7%) | 25 (12.3%) | | 3(20.0%) | 23 (11.8%) | |
| Others | 2 (2.6%) | 10 (7.5%) | | 0 (0.0%) | 12(5.9%) | | 0 (0.0%) | 12 (6.2%) | |
| **Histologic grade** | | | | | | | | | |
| Well | 33 (42.9%) | 59 (44.4%) | 0.07 | 1(16.7%) | 91 (44.6%) | 0.1 | 2 (13.3%) | 90 (46.2%) | **0.006** |
| Moderate | 40 (51.9%) | 59(44.4%) | | 5 (83.3%) | 94 (46.1%) | | 13 (86.7%) | 86 (44.1%) | |
| Poor | 4 (5.2%) | 15 (11.3%) | | 0 (0.0%) | 19 (9.3%) | | 0 (0.0%) | 19 (9.7%) | |
| **vascular invasion** | | | | | | | | | |
| Presence | 20 (26.0%) | 17 (12.8%) | **0.04** | 0 (0.0%) | 26 (12.7%) | 0.4 | 0 (0%) | 26 (13.3%) | 0.09 |
| Absence | 57 (74.0%) | 116(87.2%) | | 6 (100.0%) | 178 (87.3%) | | 15 (100%) | 169 (86.7%) | |
| **Perineural invasion** | | | | | | | | | |
| Presence | 8 (11.0%) | 14(10.5%) | 0.1 | 2 (33.3%) | 35 (17.2%) | 0.5 | 4 (26.7%) | 33 (16.9%) | 0.1 |
| Absence | 65 (89.0%) | 119(89.5%) | | 4 (66.7%) | 169 (82.8%) | | 11 (73.3%) | 162 (83.1%) | |
| **Distant metastases (M)** | | | | | | | | | |
| M0 | 42 (54.4%) | 85 (63.9%) | **0.04** | 3 (50.0%) | 83 (65.4%) | 0.3 | 8 (53.3%) | 119 (61.0%) | 0.3 |
| M1 | 35 (45.5%) | 48 (36.1%) | | 3 (50.0%) | 44 (34.6%) | | 7 (46.7%) | 76 (39.0%) | |
| **Number of metastases:** | | | | | | | | | |
| 1 | 18 (62.1%) | 33(76.7%) | 0.1 | 3 (100.0%) | 49 (70.0%) | 0.3 | 3 (40.0%) | 48 (71.6%) | 0.4 |
| ≥2 | 11 (37.9%) | 10(23.3%) | | 0 (0.0%) | 21 (30.0%) | | 2 (60.0%) | 19 (28.4%) | |
| **Removed lymph nodes:** | | | | | | | | | |
| mean ± SD | 15.0 ±9.4 | 20.0 ±9.3 | **<0.001** | 12.2 ±5.8 | 18.6 ±9.7 | **0.04** | 12.3 ±7.0 | 18.82 ±9.7 | **0.007** |
| <12 | 25 (37.9%) | 21 (15.8%) | **<0.001** | 3 (50.0%) | 43 (22.3%) | 0.1 | 6 (46.2%) | 146 (78.5%) | **0.04** |
| ≥12 | 41 (62.1%) | 112 (84.2%) | | 3 (50.0%) | 150 (77.7%) | | 7 (53.8%) | 40 (21.5%) | |
| **Positive lymph node:** | | | | | | | | | |
| Mean (± SD) | 2.4 ±3.4 | 1.4 ±3.7 | 0.07 | 0.83 ±0.4 | 1.7 ±3.6 | 0.2 | 2.3 ±3.0 | 1.7 ±3.6 | 0.9 |
| Presence | 30 (45.5%) | 34 (25.6%) | **0.003** | 5 (83.3%) | 59 (30.6%) | **0.01** | 7 (53.8%) | 57 (30.6%) | 0.08 |
| Absence | 36 (54.4%) | 99 (74.4%) | | 1 (16.7%) | 134 (69.4%) | | 6 (46.2%) | 129 (69.4%) | |
| Lymph node ratio, mean ± SD | 0.3 ±0.5 | 0.1 ±0.1 | **<0.001** | 0.4 ±0.46 | 0.2 ±0.4 | 0.1 | 0.3 ±0.4 | 0.2 ±0.4 | 0.1 |
| **Disease stages** | | | | | | | | | |
| I-II | 26 (33.8%) | 58(43.6%) | **0.04** | 1 (16.7%) | 83 (40.7%) | 0.1 | 5 (33.3%) | 79 (40.5%) | 0.3 |
| III-IV | 51 (66.2%) | 75 (56.4%) | | 5 (83.3%) | 121 (59.3%) | | 10 (66.7%) | 116 (59.5%) | |
| **BRAF mutation:** | - | - | - | - | - | - | - | - | - |
| **Phenotype MSI:** | | | | | | | | | |
| MSI | 2 (5.1%) | 15 (19.2%) | **0.03** | 0 (0.0%) | 17 (14.7%) | 0.6 | 0 (0.0%) | 17 (15.5%) | 0.3 |
| MSS | 37 (94.9%) | 63 (80.8%) | | 1 (100.0%) | 99 (85.3%) | | 7 (100.0%) | 93 (84.5%) | |
| **Follow-up time (months)** | | | | | | | | | |

*(Continued)*

**Table 6.** (Continued)

| Characteristics | KRAS mutant | KRAS wild type | p | NRAS Mutant | NRAS wild type | p | Rare mutations | wild-type status | p |
|---|---|---|---|---|---|---|---|---|---|
| **Mean SD ±** | 41.4 ±35.3 | 49.8 ±33.8 | 0.09 | 44.5 ±27.0 | 46.8 ±34.7 | 0.6 | 38.1 ±30.7 | 47.7 ±34.7 | 0.2 |
| **Recurrence** | | | | | | | | | |
| (+) | 37 (48.1%) | 38 (28.6%) | **0.02** | 2 (33.3%) | 36 (28.3%) | 0.5 | 7(46.7%) | 68 (34.9%) | 0.2 |
| (−) | 40 (51.9%) | 95 (71.4%) | | 4 (66.7%) | 91 (71.7%) | | 8 (53.3%) | 127 (65.1%) | |
| **Mortality** | | | | | | | | | |
| Death cases | 26 (34.2%) | 27 (20.3%) | **0.02** | 1(16.7%) | 73 (35.8%) | 0.6 | 3 (20.0%) | 50 (25.8%) | 0.4 |
| Censored cases | 50 (65.8%) | 106(79.7%) | | 5 (83.3%) | 131(64.2%) | | 12 (80.0%) | 144 (74.2%) | |

codon 12 mutant and codon12/13 WT CC patients (P <0.001). We documented that the codon12/13 WT status was the better predictor of prognosis. The correlation between OS and clinicopathological features was also investigated in this study. The variables including absence of distant metastases (P = 0.001), positive vascular invasion (P = 0.02), number of removed lymph node (≥12) (P = 0.01), absence of positive LN (P = 0.04), stage I-II disease (P = 0.002), and MSI phenotype (P = 0.001) revealed higher rate of OS in colon patients. The results of OS analysis are compiled in Table 8. By using Cox proportional hazards model, number of metastases (≥2), III-IV disease stage, and MSS phenotype were the independent poor prognostic factors for OS in colon cancer. However, codon12/13 WT subgroup was the better predictor factor of prognosis in CC Table 8.

## Discussion

KRAS and NRAS genes are a well-known driver oncogene in CRCs, and the presence of KRAS or NRAS mutation predicts poor response to anti-EGFR targeted therapy [8]. To our knowledge, this study is one of the first to evaluate the frequencies and clinicopathological significance of KRAS and NRAS mutations, and to analyze the prognostic impact of KRAS and NRAS mutations in patients with locally and advanced CC in the Moroccan population and the Middle East and Nord Africa region.

In this study, mutation rates of RAS, KRAS, NRAS, and BRAF genes were 39.5%, 36.7%, 2.9% %, and 0.0%, respectively. KRAS mutations were seen mainly in exon 2 (codon 12/13) (88.3% of all KRAS mutations). The frequency of KRAS mutations in this cohort is in accordance with that in previous reports, which reported that approximately 34~45% of the patients had KRAS mutations [16,17].While, the rate of RAS, BRAF and KRAS exon 2 mutations was slightly lower than that in two recently published studies [12,15]. Previous studies have reported that the most common KRAS mutation types are p.Gly12Asp, p.Gly12Val and p.Gly13Asp. In our cohort, the most common single mutation identified was a G>A transition (c.35G>A, p.G12D) in codon 12 of exon 2 (27.9%), followed by the G12V mutation (25.0%), this finding is generally consistent with data found in the literature [18,19]. Furthermore, our findings revealed a lower rate of p.G13D (c.38 G>A) mutation in codon 13 (19.1%) compared to the rate of G12D and G12V mutations in codon 12. This result is in agreement with that reported by NIU *et al* [20] and Omidifar *et al* [21].

We detected 7.14% of rare mutations (KRAS exons 3 and 4 and NRAS exons 2, 3, and 4) in this study. These mutations were distributed between KRAS exon 3 mutants (0.5%), KRAS exon 4 mutants (2.4%), and NRAS mutants (exons 2 and 3) (2.9%), we did not find any NRAS mutation in exon 4. The prevalence of rare mutations in our cohort is higher than that reported by Rimbert et al recently (15) and other studies [17,22]. There are some explanations

**Table 7. KRAS exon 2 mutation subtypes and clinicopathological features.**

| Characteristics | KRAS codon 12 mutant | KRAS codon 12 wild type | p | Kras codon 13 mutants | KRAS codon 13 wild type | p | KRAS codon 12 mutants | KRAS codon 13 mutants | p |
|---|---|---|---|---|---|---|---|---|---|
| **Age** | | | | | | | | | |
| Mean ((±SD) | 57.2 ± 14.0 | 55.0 ±15.1 | 0.3 | 57.3±20.1 | 55.4±14.3 | 0.6 | 57.2 ± 14.0 | 57.3±20.1 | 0.9 |
| ≤55 | 22 (42.3%) | 70 (44.3%) | 0.4 | 6 (37.5%) | 86 (44.3%) | 0.1 | 22 (42.3%) | 6 (37.5%) | 0.2 |
| >55 | 30 (57.7% | 88 (55.7%) | | 10 (62.5%) | 108 (55.7%) | | 30 (57.7%) | 10 (62.5%) | |
| **Gender** | | | | | | | | | |
| Female | 31(59.6%) | 66 (41.8%) | **0.01** | 10 (62.5%) | 87 (44.8%) | 0.08 | 31(59.6%) | 10 (62.5%) | 0.2 |
| Male | 21(40.4%) | 92 (58.2%) | | 6 (37.5%) | 107 (55.2%) | | 21(40.4%) | 6 (37.5%) | |
| **Tumor site** | | | | | | | | | |
| Right colon | 14 (26.9%) | 53(33.5%) | 0.2 | 3 (18.8%) | 64 (33.0%) | 0.1 | 14 (26.9%) | 3 (18.8%) | 0.09 |
| Left colon | 38 (73.1%) | 105(66.5%) | | 13 (81.3%) | 130 (67.0%) | | 38(73.1%) | 13 (81.3%) | |
| **Histologic subtype** | | | | | | | | | |
| Adenocarcinoma | 43 (82.7%) | 133(84.2%) | **0.02** | 14(87.5%) | 158 (81.4%) | 0.1 | 43 (82.7%) | 14(87.5%) | 0.1 |
| Mucinous adenocarcinoma | 7 (13.5%) | 15 (9.5%) | | 2(12.5%) | 24 (12.4%) | | 7 (13.5%) | 2(12.5%) | |
| Others | 2 (3.8%) | 10 (6.3%) | | 0 (0.0%) | 12 (6.2%) | | 2 (3.8%) | 0 (0.0%) | |
| **Histologic grade** | | | | | | | | | |
| Well | 24 (46.2%) | 68 (43.0%) | 0.8 | 8(50.0%) | 84 (43.3%) | 0.1 | 24 (46.2%) | 8(50.0%) | **0.1** |
| Moderate | 24 (46.2%) | 75 (47.5%) | | 8 (50.0%) | 91 (46.9%) | | 24 (46.2%) | 8 (50.0%) | |
| Poor | 4 (7.7%) | 15 (9.5%) | | 0 (0.0%) | 19 (9.8%) | | 4 (7.7%) | 0 (0.0%) | |
| **Vascular invasion** | | | | | | | | | |
| Presence | 15 (28.8%) | 22 (13.9%) | **0.01** | 3 (18.8%) | 34 (17.5%) | 0.1 | 8 (16.3%) | 2 (12.5%) | 0.1 |
| Absence | 37 (71.2%) | 136(86.1%) | | 13(81.3%) | 160 (82.5%) | | 41 (83.7%) | 14 (87.5%) | |
| **Perineural invasion** | | | | | | | | | |
| Presence | 7 (14.3%) | 15 (9.6%) | 0.2 | 0 (0.0%) | 26 (13.4%) | 0.1 | 7 (14.3%) | 0 (0.0%) | 0.1 |
| Absence | 42 (85.7%) | 142(90.4%) | | 16 (100.0%) | 168 (86.6%) | | 42 (85.7%) | 16 (100.0%) | |
| **Distant metastases (M)** | | | | | | | | | |
| M0 | 29 (55.8%) | 98 (62.0%) | 0.2 | 8 (50.0%) | 119 (61.3%) | 0.1 | 29 (55.8%) | 8 (50.0%) | 0.2 |
| M1 | 23 (44.2%) | 60 (38.0%) | | 8 (50.0%) | 75 (38.7%) | | 23 (44.2% | 8 (50.0%) | |
| **Number of metastases:** | | | | | | | | | |
| 1 | 15 (71.4%) | 36 (70.6%) | 0.5 | 2 (33.3%) | 49 (73.1%) | 0.1 | 15 (71.4%) | 2 (33.3%) | 0.08 |
| ≥2 | 6 (28.6%) | 15 (29.4%) | | 4 (66.7%) | 18 (26.9%) | | 6 (28.6%) | 4 (66.7%) | |
| **Removed lymph nodes:** | | | | | | | | | |
| mean ± SD | 15.6±10.1 | 20.4 ±9.3 | <**0.001** | 14.6 ±7.9 | 18.7 ±9.7 | **0.05** | 15.6±10.1 | 14.6 ±7.9 | 0.7 |
| <12 | 16 (36.4%) | 30 (19.4%) | **0.01** | 6 (40.0%) | 40(21.7%) | **0.05** | 16 (36.4%) | 6 (40.0%) | 0.2 |
| ≥12 | 28 (63.6%) | 125 (80.6%) | | 9 (60.0%) | 144 (78.3%) | | 28 (63.6%) | 9 (60.0%) | |
| **Positive lymph node:** | | | | | | | | | |
| Mean (± SD) | 2.9 ±3.7 | 1.4 ±3.5 | **0.02** | 1.3 ±2.4 | 1.7 ±3.7 | **0.5** | 2.9 ±3.7 | 1.3 ±2.4 | **0.01** |
| Presence | 22 (50.0%) | 42 (27.1%) | **0.004** | 6 (40.0%) | 58 (31.5%) | 0.1 | 22 (50.0%) | 6 (40.0%) | **0.1** |
| Absence | 22 (50.0%) | 113(72.9%) | | 9 (60.0%) | 126 (68.5%) | | 22 (50.0%) | 9 (60.0%) | |
| Lymph node ratio, mean ± SD | 0.4±0.6 | 0.1 ±0.2 | <**0.001** | 0.2 ±0.3 | 0.1 ±0.4 | 0.9 | 0.4±0.6 | 0.2 ±0.3 | **0.02** |
| Disease stages | | | | | | | | | |
| I-II | 15 (28.8%) | 69 (43.7%) | **0.04** | 7 (43.8%) | 77 (39.7%) | 0.1 | 15 (28.8%) | 7 (43.8%) | 0.2 |
| III-IV | 37 (71.2%) | 89 (56.3%) | | 9 (56.3%) | 117 (60.3%) | | 37 (71.2%) | 9 (56.3%) | |
| BRAF mutation: | - | - | - | - | - | - | - | - | - |

*(Continued)*

**Table 7.** (Continued)

| Characteristics | KRAS codon 12 mutant | KRAS codon 12 wild type | p | Kras codon 13 mutants | KRAS codon 13 wild type | p | KRAS codon 12 mutants | KRAS codon 13 mutants | p |
|---|---|---|---|---|---|---|---|---|---|
| **Phenotype MSI:** | | | | | | | | | |
| MSI | 2 (8.0%) | 15 (16.3%) | 0.2 | 0 (0.0%) | 17 (15.6%) | 0.2 | | | |
| MSS | 23 (92.0%) | 77 (83.7%) | | 8 (100.0%) | 92 (84.4%) | | | | |
| **Follow-up time (months)** | | | | | | | | | |
| **Mean SD ±** | 46.6 ±37.8 | 46.7 ±33.5 | 0.9 | 29.0 ±23.3 | 48.0 ±34.9 | **0.006** | 46.6 ±37.8 | 29.0 ±23.3 | **<0.001** |
| **Recurrence** | | | | | | | | | |
| (+) | 25 (48.1%) | 50 (31.6%) | **0.02** | 7 (43.8%) | 68(35.1%) | 0.1 | 25 (48.1%) | 7 (43.8%) | 0.2 |
| (−) | 27 (51.9%) | 108(68.4%) | | 9 (56.3%) | 126 (64.9%) | | 27 (51.9%) | 9 (56.3%) | |
| **Mortality** | | | | | | | | | |
| Death cases | 15 (28.8%) | 38 (24.1%) | 0.2 | 9(56.3%) | 44 (22.8%) | **0.003** | 15 (28.8%) | 9(56.3%) | **0.03** |
| Censored cases | 37 (71.2%) | 120(75.9%) | | 7 (43.8%) | 149 (77.2%) | | 37 (71.2%) | 7 (43.8%) | |

for this difference. First, we enrolled both the locally and advanced CC patients, second, we analyzed the RAS status using Pyrosequencing technology, which is considered to show higher sensitivity (5%) than that of other testing methods. A study conducted by Tougeron et al [23], demonstrated that Pyrosequencing detected 17.9% of the KRAS mutations in patients with KRAS wild-type using direct sequencing alone.

NRAS mutations were identified in 6 (2.9%) out of 210 tumor samples. This incidence is in concordance with what was reported in previously published data, which concluded that NRAS mutations occur in 2–7% of cases [8,24]. The mutation frequency of NRAS exon 3 (4, 2.0%) was higher than in NRAS exon 2 (2, 1.0%). These findings are similar to Guo's study of 353 Chinese colorectal cancer patients [25]. NRAS mutations can coexist with KRAS mutations [26], in our analysis; we did not find any case with simultaneous mutations. In our study, from 200 patients, we did not find any BRAF mutation. This result confirms the rarity of the V600E mutation in colon cancer, epically in the Moroccan population [27].

This study demonstrates that RAS and KRAS mutant status underline special clinicopathological and molecular features. As reported in some studies [15,28,29], in our context, RAS mutated CC was associated with female gender, distal colon, classical adenocarcinoma subtype, moderately differentiated tumors, and presence of vascular invasion. Interestingly, our study demonstrated that CC with RAS and epically KRAS mutations show a lower rate of the total lymph node. Recently, NOGUERA et al [29] found the same observation, but the difference was not statistically significant. Besides, patients with KRAS mutations showed a significantly higher rate of positive lymph nodes. This finding is in line with the results reported by Mannan et Hahn-Stro [30] and an earlier study by Oliveira et al [31]. In the Chinese group, previous reports suggest that KRAS mutated CCs have no specific trend in lymph node metastasis [32].

Regarding the correlation between KRAS mutations and tumor site, there is some dissimilarity in the literature. Some studies have shown that KRAS mutations occur more frequently in the right colon [32,33]. In contrast and similar to our analysis, Kawazoe et al [17] demonstrate that distal colon harbors more KRAS mutations. In the same way, some studies [15,34] have shown that KRAS mutations tended to occur more frequently in classical adenocarcinoma tumors as reported in our study. However, other has observed an increased tendency of these mutations in the mucinous adenocarcinoma subtype [32]. Significance association between KRAS mutations and advanced TNM stage was also observed in our study, which further supports previous findings [30]. Although, Zhang et al [32] did not find any association.

**Table 8. The molecular and clinical variables associated with overall survivals in the 210 colon cancer.**

| Variables | Univariate analysis | | Multivariate analysis | |
|---|---|---|---|---|
| | Mean OS months (95% CI) | P value | Hazard ratio (95% CI) | P value |
| Age (yr) | | | | |
| ≥55 | 95.52(84.85–106.18) | 0.9 | | |
| <55 | 96.03(83.19–108.88) | | | |
| Gender | | | | |
| Female | 91.84 (78.98–104.70) | 0.4 | | |
| Male | 98.09 (87.26–108.92) | | | |
| Tumor site | | | | |
| Right side | 93.75 (79.58–107.92) | 0.8 | | |
| Left side | 90.34 (81.79–98.90) | | | |
| Histologic subtype | | | | |
| Adenocarcinoma | 95.42 (86.08–104.76) | 0. 3 | | |
| Mucinous adenocarcinoma | 94.11 (74.50–113.73) | | | |
| Others | 61.33 (37.33–85.33) | | | |
| Histologic grad | | | | |
| Well | 96.23 (84.53–107.94) | 0.6 | | |
| Moderate | 96.65 (84.05–109.25) | | | |
| Poor | 73.45 (51.38–95.52) | | | |
| Distant metastases | | | | |
| M0 | 105.10 (96.10–114.10) | **0.001** | 0.34 (0.61–2.77) | 0.4 |
| M1 | 75.33 (59.12–91.54) | | | |
| Number of metastases: | | | | |
| 1 | 50.12 (34.71–65.54) | 0.1 | 1.26 (0.05–1.05) | 0.05 |
| ≥2 | 84.45 (56.81–112.09) | | | |
| Tumor stage | | | | |
| I-II | 109.41(99.52–119.51) | **0.002** | 1.98 (0.00–0.21) | **0.003** |
| III-IV | 84.63 (72.57–96.69) | | | |
| TLN | | | | |
| ≥12 | 102.26 (93.41–111.10) | **0.01** | 0.66 (0.29–3.32) | 0.9 |
| <12 | 72.51 (57.02–88.01) | | | |
| Positive LN | | | | |
| Presence | 80.54 (66.88–94.19) | **0.04** | 0.31 (0.46–4.15) | 0.5 |
| Absence | 102.09 (92.71–111.47 | | | |
| Perineural invasion | | | | |
| Yes | 91.56 (77.95–105.18) | 0.2 | | |
| No | 94.97 (86.34–103.60) | | | |
| Vascular invasion | | | | |
| Yes | 102.23 (90.81–113.66) | **0.02** | 1.93 (0.20–20.78) | 0.5 |
| No | 92.25 (83.25–101.25) | | | |
| MSI status | | | | 0.05 |
| MSI | 109.71 (99.85–119.58) | **0.001** | 0.11 (0.06–1.04) | |
| MSS | 74.08 (68.00–80.16) | | | |
| Full RAS status | 104.29 (94.35–114.02) | **0.006** | | |
| Wild-type | | | 0.33 (0.00–4.50) | 0.9 |
| Mutant | 78.18 (65.84–90.52) | | | |
| KRAS status | | | | |
| Wild-type | 104.14 (94.48–113.80) | **0.003** | | |

*(Continued)*

**Table 8.** (Continued)

| Variables | Univariate analysis | | Multivariate analysis | |
|---|---|---|---|---|
| | Mean OS months (95% CI) | P value | Hazard ratio (95% CI) | P value |
| Mutant | 76.81 (63.89–89.74) | | 0.29 (0.00–3.01) | 0.9 |
| NRAS status | | | | |
| Wild-type | 95.68 (87.36–104.01) | 0.6 | | |
| Mutant | 73.00 (48.99–97.00) | | | |
| KRAS exons 2 codon: | | | | |
| codon 12/13 WT | 103.75 (94.32–113.17) | <**0.001** | | |
| codon 12 mutant | 84.56 (69.29–99.82) | | 0.35 (0.82–14.08) | **0.01** |
| codon 13 mutant | 40.64 (23.02–58.27) | | | |

Interestingly, we documented a positive relationship between KRAS mutated CC and microsatellite stability phenotype, which is consistent with findings reported by Rimbert et al [15,35]. There was no significant association between age and KRAS mutations in our analysis, which is similar to previous studies [17,32].

In this study, we also compared the clinicopathological features between codon 12 CC and codon 13-mutated CC. As a point of interest, the result indicates that codon 13 gene mutation was associated with a higher number of death cases, a lower rate of positive lymph node, and a lower rate of LNR. These findings suggest that tumors with codon 13 mutations are more aggressive than in codon 12. Recently, the molecular epidemiological study conducted by Gonsalves et al [36] has shown that KRAS codon 13 mutations were associated with deficient mismatch repair phenotype and poor differentiation. These parameters were reported to be predictors of worse OS [37]. These results may explain the worse survival observed in patients with codon 13 mutations CC, as reported by Modest et al [38]. However, Mannan et al [39] reported have that KRAS codon 12-mutated CC is more aggressive than codon 13-mutated CC because they were associated with factors of poor prognosis like advanced tumor staging (p = 0.02) and nodular metastasis (p = 0.04).

Another remarkable investigation in this study is the correlation between a patient's clinicopathological features and rare KRAS and NRAS mutations (KRAS exons 3 and 4 and NRAS mutations). We documented that, in comparison with wild-type tumors, rare mutations tumors were more likely to exhibit moderate differentiation, lower lymph node rate, and lower incidence of ≥12 lymph nodes. Moreover, RAS rare-mutated tumors tended to be associated with the left tumor site (*P = 0.08*) and higher lymph node metastasis rate (*P = 0.08*) with a difference close to significance. To the best of our knowledge, we are the first to illustrate the clinicopathological and survival differences between rare RAS mutated CC and wild-type CC in the Middle East and Nord Africa (MENA) region. While the publication of the Molecular Cancer Genetics Platform of Poitiers (French National Cancer Institute (INC)) was the first world-wild detailed report concerning the association between clinicopathological features and rare mutation status. The investigators of this study [15] have demonstrated that KRAS exon 3-mutated CC and KRAS exon 4-mutated CC were both associated with mucinous histological subtypes. Additionally, KRAS exon 3-mutated CC tended to be correlated with the rectal tumor site.

Finally, we found that in comparison with wild-type RAS status, NRAS mutated status was significantly associated with a lower lymph node rate and a higher lymph node metastasis rate. A recent Chinese study on Chinese patients showed that NRAS mutations occurred more frequently in female patients [25]. Others report did not find any correlations between NRAS

mutations and clinicopathological characteristics [15,32]. This discrepancy of results may be attributed to the sampling size included in each study, the rarity of NRAS mutations in CC, and few available data concerning the association between NRAS mutations and clinicopathological features.

Regarding our second objective, our analysis is the first to evaluate the relationship between KRAS and NRAS mutations and clinical outcome in CC patients, in the Middle East and North Africa region. We found that KRAS and NRAS mutations appear to have worse OS in comparison to RAS wild-type in CC patients at all stages. The same results were reported by Yaeger et al [40] in 2015. We also demonstrated that KRAS mutations exhibit similar behavior to full RAS mutations concerning OS, while KRAS wt patients show an improved OS. In the literature, the prognostic role of KRAS mutations as a global prognostic biomarker factor of OS remains controversial. Several studies have reported a statistically significant reduction of OS in the presence of KRAS mutations [41–43] as reported in our analysis. In contrast, Ogino et al [44] did not identify a relevant prognostic role of KRAS mutations in patients with CC. Furthermore, no association between KRAS mutations and survival was found in the PETACC-3 trial among 1404 colon cancer patients [45].

In our study, we epically determined and compared the impact on OS between three groups of tumors: tumors with KRAS codon 13 mutations, tumors with KRAS codon 12 mutations, and KRAS codon 12/13 wild type tumors. We documented that tumors carried KRAS codon 13 mutations show the poorer and adverse OS as compared to those with KRAS codon 12 mutations. Although, KRAS codon 12/13 wild-type tumors had shown better OS between the three groups (P<0.001). In the literature, a few subsequent clinical studies and meta-analyses have been conducted on CRC patients for this issue [46,47]. Recently, Kwak and colleagues have illustrated in a systematic review that KRAScodon13 gene mutation appears to have worse OS in comparison to KRAS wild-type in CRC patients but shows similar clinical outcomes to codon 12 gene mutations [48]. In contrast, Imamura et al and Taieb et al [49] showed that KRAS codon 12 mutations are associated with inferior survival in comparison with mutations at codon 13 [50]. NRAS mutations seemed to have any clinical impact on OS in our cohort. Considering we found only 6 patients harboring NRAS mutations, this finding does not yet allow us to draw a firm conclusion of the prognostic value of NRAS status. Recent clinical studies were also limited by a small number of NRAS-mutant cases to establish their clinical effects on survival [51]. In contrast, a case series from Italy of mCRC patients that included 47 cases with NRAS mutant disease found a significant correlation between NRAS mutation and shorter OS compared with wild-type cases [52]. Additionally, Wang et al documented that in a retrospective analysis of stage I–IV colorectal cancers, the presence of an NRAS mutation was associated with significantly shorter survival [53].

## Conclusion

Our study demonstrates that KRAS and NRAS mutations occur respectively in 36.7% and 2.9% of our patients. Our findings show that KRAS and NRAS mutations especially KRAS mutations have distinct clinicopathological features (tumor site, MSS status, positive lymph node. . .). Among KRAS and NRAS mutations, the KRAS codon 13-mutated status was the worst predictor of prognosis at all stages in our population.

## Author Contributions

**Conceptualization:** Sanae el Bardai.

**Data curation:** Sanae el Bardai, Zineb Benbrahim, Laila Chbani.

**Formal analysis:** Ihsane El Otmani.

**Funding acquisition:** Laila Chbani.

**Investigation:** Sidi Adil Ibrahimi, Laila Chbani.

**Methodology:** Fatima El agy, Sanae el Bardai, Laila Chbani.

**Project administration:** Laila Chbani.

**Resources:** Zineb Benbrahim, Ibn Majdoub Hassani Karim, Khalid Mazaz, El Bachir Benjelloun, Abdelmalek Ousadden, Mohammed El Abkari, Sidi Adil Ibrahimi.

**Software:** Ihsane El Otmani.

**Validation:** Laila Chbani.

**Writing – original draft:** Fatima El agy.

**Writing – review & editing:** Fatima El agy, Laila Chbani.

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
