## [Decision Letter · Decision Letter 0]

10 Dec 2020

PONE-D-20-28747

Mutation status and prognostic value of KRAS and NRAS mutations in Moroccan colon cancer patients: a first report.

PLOS ONE

Dear Dr. Fatima

 The reviewers raised major concerns including: How MIS of CRC established, why an unmethylated control used in Pyrosequencing, since the study is not about methylation?

there different NRAS mutations levels. It should be consistence. 

Thank you for submitting your manuscript to PLOS ONE. After careful consideration, we feel that it has merit but does not fully meet PLOS ONE’s publication criteria as it currently stands. Therefore, we invite you to submit a revised version of the manuscript that addresses the points raised during the review process.

We look forward to receiving your revised manuscript.

Kind regards,

Hassan Ashktorab

Academic Editor

PLOS ONE

Journal Requirements:

2. Our journal requires that methods are described in enough detail to allow suitably skilled investigators to fully replicate your study. If materials, methods, and protocols are well established, authors may cite articles where those protocols are described in detail, but the submission should include sufficient information to be understood independent of these references. Please revise your manuscript so that you methodology for BRAF molecular testing is briefly but sufficiently described.

In addition, please provide the primer sequences for all primers used.

Also, please change p-values of "0.000" to "<0.001" in your tables.

3. Please also provide additional information about the participant recruitment method and the demographic details of your participants. Please ensure you have provided sufficient details to replicate the analyses such as:

a) a description of any inclusion/exclusion criteria that were applied to participant recruitment,

b) a statement as to whether your sample can be considered representative of a larger population, and

c) a description of how participants were recruited.

4. Please provide additional details regarding participant consent.

In the ethics statement in the Methods and online submission information, please ensure that you have specified what type you obtained (for instance, written or verbal, and if verbal, how it was documented and witnessed). If your study included minors, state whether you obtained consent from parents or guardians. If the need for consent was waived by the ethics committee, please include this information.

6. Thank you for stating the following financial disclosure:

'The funders had no role in study design, data collection and analysis, decision to publish, or preparation of the manuscript.'

Reviewers' comments:

Reviewer's Responses to Questions

**Comments to the Author**

1. Is the manuscript technically sound, and do the data support the conclusions?

Reviewer #1: Yes

Reviewer #2: Yes

2. Has the statistical analysis been performed appropriately and rigorously? 

Reviewer #1: Yes

Reviewer #2: Yes

3. Have the authors made all data underlying the findings in their manuscript fully available?

Reviewer #1: Yes

Reviewer #2: Yes

4. Is the manuscript presented in an intelligible fashion and written in standard English?

Reviewer #1: Yes

Reviewer #2: No

5. Review Comments to the Author

Reviewer #1: This paper describes RAS mutations in Moroccan CRC patients (210). While the data is sound and confirms previous findings from other populations, this is the first of its kind in Morrocan patients. Changes need to be made to improve the manuscript. The Abstract is too long and the Discussion needs some revision.

Why is it that an unmethylated control used in Purosequencing, this study is not about methylation?

The authors need to explain how was MSI status established, It is NOT i Methods section?

In Methods , they stated they compared stges I-II vs II-IV, I think you mean I-II vs. III-IV, Pease correct.

What is most important is Positive and examined Lymph nodes, NOT harvested/Collected LN. Please adjust write up to show that mutant Kras generally needed LESS LN to be examined to get positve ones.

For NRAS mutations , is it 2-8 or 2-9%, you are putting different numbers in different places?

Would suggest to drop RAS muataions part from text and directly just state KRAS and NRAS.

I tables, please bolden and underline SIGNIFICANT p VALUES.

The paper also uses many unusual abbreviations amd has many typos and grammatical errors. Please review completely.

Reviewer #2: There is a slight redundancy in the Introduction such as “Ras-family G-proteins transduce signals in growth factor signals, such as EGF receptor (EGFR) signaling pathway (1). I suggest you strike out “signals” that appears before “in growth factor”.

The citation numbers need a single space separation from the words. Make space between the last words before cited reference numbers in your text.

Your statement that “KRAS mutations occur in 45% of colorectal cancer (CRC) and NRAS in only 5-8%(6)” should be clarified to indicate whether this is applicable to the Moroccan patients or is referring to the world-wide CRC patients.

-Essentially, the authors of this article are attempting to mimic the work of the French authors in Molecular Cancer Genetics Platform of Poitiers (French), who have published their results about the clinicopathological characteristics of the RAS mutations in a cohort of over 1500 colorectal cancer patients. Was the French study done exclusively with French CRC patients?

-Table 6. Any reason why the multivariate analysis was not done in some of the metastatic tumors in Table 6, where the molecular and clinical variables associated with overall survivals in the 210 colon cancer patients were shown?

-In the body of the text, especially in the Results section, it was quite difficult to follow the different mutational hotspots from gene site to gene site, it would have been better to display the most common types in tabular format for quicker views and understanding of the cardinal gene alterations.

-The impetus to investigate the WT and mutated RAS gene sites was that mutations in KRAS outside exon 2 (KRAS exons 3,4) and in other downstream effectors of the EGFR signaling pathway (like NRAS) were published to be associated with resistance to anti-EGFR monoclonal antibodies that were used as a standard regimen for CRC patients – meaning that the two Anti-EGFR monoclonal antibodies, panitumumab and cetuximab, used to treat CRC patients with KRAS and NRAS mutations, were not effective. Therefore, the policy was to successfully treat CRC patients, the full RAS WT status had to be confirmed in all mCRC patients before initiating treatment. The question is: Does this study conclusively resolve that issue? If so, that must be clearly stated in the manuscript.

-Compared to NRAS wild-type patients, “NRAS-mutated patients seem to be exhibiting lower total lymph node rate, and were associated with higher lymph node metastasis number”. This statement is confusing. It could be revised to make the right statement without seemingly contradicting itself. Among RAS mutations, KRAS codon 13 mutations seemed to have a poor prognostic value, which may negatively impinge on overall survival of all the stages of CRC patients. Is that a correct statement?

6. PLOS authors have the option to publish the peer review history of their article (what does this mean?). If published, this will include your full peer review and any attached files.

Reviewer #1: No

Reviewer #2: No

---

## [Author Response · Author response to Decision Letter 0]

18 Jan 2021

1. Please ensure that your manuscript meets PLOS ONE's style requirements, including those for file naming:

Response: this point has been verified.

2. Our journal requires that methods are described in enough detail to allow suitably skilled investigators to fully replicate your study. If materials, methods, and protocols are well established, authors may cite articles where those protocols are described in detail, but the submission should include sufficient information to be understood independent of these references. Please revise your manuscript so that you methodology for BRAF molecular testing is briefly but sufficiently described.

In addition, please provide the primer sequences for all primers used.

Also, please change p-values of "0.000" to "<0.001" in your tables.

Response: the methodology of BRAF molecular testing has been briefly and sufficiently described “BRAF testing was performed for 200 patients using Sanger sequencing as described previously (KRAS and NRAS mutations testing). Briefly, DNA was amplified by PCR (Master Mix (2X) kits) according to the manufacturer’s protocols and using the following forward primer (5’-ACGAACGAGACTATCCTTTTAC-3’) and reverse primer (5’- CATTGAGTCGTCGTAGAGTCCC -3’). PCR products were purified using ExoSAP® kit according to the manufacturer’s protocols. The purified PCR products were sequenced using the direct sequencing with BigDye Terminator V3.1 Cycle Sequencing Kit (ABI Prism) and analyzed on Applied Biosystems 3500Dx Genetic Analyzer (Applied Biosystems).”

The primer sequences used for PCR have been added in methods section:

Table 1: Primer sequences used for PCR.

Primer name Primer sequence 

KRAS-ex 2- F 

KRAS-ex 2- R 

5’-GGTGGAGTATTTGATAGTGTA- 3’

5’-TGCATATTACTGGTGCAGACC- 3’

KRAS-ex 3- F 

KRAS-ex 3- R 5’-AGTAAAAGGTGCACTGTAATAA-3’

5’-ATAATAAGCTGACATTAAGGAG-3’

KRAS-ex 4- F

KRAS-ex 4- R

 5’-TGTTACTAATGACTGTGCTATAACTTTT-3’

5’-TATGCTATACTATACTAGGAAATAAAA-3’

NRAS-ex2-F

NRAS-ex2-R 5’-ATGACTGAGTACAAACTGGTGGTGGTTGGAGCA-3’

5’-CACTTTGTAGATGAATATGATCCCACCATAGAG-3’

NRAS-ex3-F

NRAS-ex3-R 5’-GATTCTTACAGAAAACAAGTGGTTA-3’

5’-CATTTGCGGATATTAACCTCTACAG-3’

NRAS-ex4-F

NRAS-ex4-R 5’-GGAGCAGATTAAGCGAG-3’

5’-TCAGCCAAGACCAGACAG-3’

The p-values of "0.000» have been changed to "<0.001" in all tables.

3. Please also provide additional information about the participant recruitment method and the demographic details of your participants. Please ensure you have provided sufficient details to replicate the analyses such as:

a) a description of any inclusion/exclusion criteria that were applied to participant recruitment,

b) a statement as to whether your sample can be considered representative of a larger population, and

c) a description of how participants were recruited.

Response: This point has been addressed in methods section. Medical charts were prospectively and retrospectively reviewed, and patients were recruited using the following selection criterion: (a) patients with histologically confirmed primary adenocarcinoma (b) all cases with pathologic I-IV stage colon cancer, (c) patients with prognostic information and who underwent surgical resection for CC tumor at the surgery department of Hassan II University Hospital. Patients were excluded if their records were incomplete, without histological confirmation of colon adenocarcinoma and if they had rectal cancer. Demographic and clinicopathological information (e.g. age, gender, tumor grade, tumor site histological subtype, disease stage, and numbers of examined regional lymph nodes…) and follow up data, were obtained from the patient’s medical records and pathology reports. Overall survival was defined as the time from the start of diagnosis until death or until the last follow-up.

4. Please provide additional details regarding participant consent.

In the ethics statement in the Methods and online submission information, please ensure that you have specified what type you obtained (for instance, written or verbal, and if verbal, how it was documented and witnessed). If your study included minors, state whether you obtained consent from parents or guardians. If the need for consent was waived by the ethics committee, please include this information

Response: This study protocol was reviewed and approved by Hassan II University Hospital Ethics Committee of FEZ, Morocco, under reference no. 13/18. All patients gave written informed consent before the start of the study.

Response: this point has been addressed. Data Availability Statement: All relevant data are within the Manuscript.

6. Thank you for stating the following financial disclosure:

'The funders had no role in study design, data collection and analysis, decision to publish, or preparation of the manuscript.'

a. Please clarify the sources of funding (financial or material support) for your study. List the grants or organizations that supported your study, including funding received from your institution.

d. If you did not receive any funding for this study, please state: “The authors received no specific funding for this work.”

Response: this point has been addressed. Funding: The authors received no specific funding for this work.

Response to review comments:

Reviewer #1:

This paper describes RAS mutations in Moroccan CRC patients (210). While the data is sound and confirms previous findings from other populations, this is the first of its kind in Morrocan patients. Changes need to be made to improve the manuscript. The Abstract is too long and the Discussion needs some revision.

Response: this point has been addressed in the manuscript. 

Abstract: This study aimed to estimate the incidence of KRAS, NRAS, and BRAF mutations in the Moroccan population, and investigate the associations of KRAS and NRAS gene mutations with clinicopathological characteristics and their prognosis value. To achieve these objectives, we reviewed medical and pathology reports for 210 patients. RAS testing was investigated by Sanger sequencing and Pyrosequencing technology. BRAF (exon 15) status was analyzed by the Sanger method. The expression of MMR proteins was evaluated by Immunohistochemistry. KRAS and NRAS mutations were found in 36.7% and 2.9% of 210 patients, respectively. KRAS exon 2 mutations were identified in 76.5% of the cases. RAS-mutated colon cancers were significantly associated with female gender, presence of vascular invasion, classical adenocarcinoma, moderately differentiated tumors, advanced TNM stage III-IV, left colon site, higher incidence of distant metastases at the time of diagnostic, microsatellite stable phenotype, lower number of total lymph nodes, and higher means of positive lymph nodes and lymph node ratio. KRAS exon 2-mutated colon cancers, compared with KRAS wild-type colon cancers were associated with the same clinicopathological features of RAS-mutated colon cancers. NRAS-mutated patients were associated with lower total lymph node rate and the presence of positive lymph node. Rare RAS-mutated tumors, compared with wild-type tumors were more frequently moderately differentiated and associated with lower lymph node rate. We found that KRAS codon 13-mutated, tumors compared to codon 12-mutated tumors were significantly correlated with a higher death cases number, a lower rate of positive lymph, lower follow-up time, and poor overall survival. Our findings show that KRAS and NRAS mutations have distinct clinicopathological features. KRAS codon 13-mutated status was the worst predictor of prognosis at all stages in our population.

Why is it that an unmethylated control used in Purosequencing, this study is not about methylation?

Response: this positive control has been designed by the supplier for use in both methylation and PCR testing. 

The authors need to explain how was MSI status established, it is NOT in Methods section?

Response: The mismatch repair tumor status (MSS or MSI) was established by immunohistochemistry (IHC) to detect the intact or the loss expression of the MMR proteins (MLH1, PMS2, MSH2, and MSH6). The IHC study was assessed on unstained formalin-fixed paraffin-embedded (FFPE) tumor tissue sections of 4 μm thickness, on the automated immunostainer Ventana Benchmark ULTRA. We have employed monoclonal antibodies specific for each MMR protein, MLH1 (G168-728/CELL MARQUE), MSH2 (G219-1129/CELL MARQUE), MSH6 (44/CELL MARQUE), and PMS2 (MRQ-28/CELL MARQUE). Adjacent normal tissue (lymphocytes or normal glandular cells) was used as an internal control for positive staining.

In Methods, they stated they compared stges I-II vs II-IV, I think you mean I-II vs. III-IV, Pease correct.

Response: This point has been corrected.

Please adjust write up to show that mutant Kras generally needed LESS LN to be examined to get positve ones.

Response: Furthermore the presence of positive lymph node was significantly associated with KRAS mutated tumors.

For NRAS mutations, is it 2-8 or 2-9%, you are putting different numbers in different places?

Response: This point has been corrected in the text; it is 2.9%.

Would suggest to drop RAS muataions part from text and directly just state KRAS and NRAS.

Response: this point has been addressed in the text.

In tables, please bolden and underline SIGNIFICANT p VALUES.

 Response: this point has been addressed in all tables

The paper also uses many unusual abbreviations amd has many typos and grammatical errors. Please review completely.

Response: This point has been checked.

Reviewer #2: 

There is a slight redundancy in the Introduction such as “Ras-family G-proteins transduce signals in growth factor signals, such as EGF receptor (EGFR) signaling pathway (1). I suggest you strike out “signals” that appears before “in growth factor”.

Response: Ras-family G-proteins transduce in growth factor signals, such as EGF receptor (EGFR) signaling pathway.

The citation numbers need a single space separation from the words. Make space between the last words before cited reference numbers in your text.

Response: this point has been addressed.

Your statement that “KRAS mutations occur in 45% of colorectal cancer (CRC) and NRAS in only 5-8%(6)” should be clarified to indicate whether this is applicable to the Moroccan patients or is referring to the world-wide CRC patients.

Response: Numerous studies have reported that KRAS and NRAS mutations occur respectively in 45% and 5-8% of worldwide colorectal cancer patients (6).

-Essentially, the authors of this article are attempting to mimic the work of the French authors in Molecular Cancer Genetics Platform of Poitiers (French), who have published their results about the clinicopathological characteristics of the RAS mutations in a cohort of over 1500 colorectal cancer patients. Was the French study done exclusively with French CRC patients?

Response: Recently, the Molecular Cancer Genetics Platform of Poitiers (French) has published interesting results about the clinicopathological features of every KRAS and NRAS mutation in a cohort of 1735 French colorectal cancer patients, enrolled from 28 hospital molecular genetics platforms throughout France (15).

-Table 6. Any reason why the multivariate analysis was not done in some of the metastatic tumors in Table 6, where the molecular and clinical variables associated with overall survivals in the 210 colon cancer patients were shown?

Response: The multivariate analysis was performed; we just forgot to present the results in table 6. 0.34 (0.61-2.77) 0.4

-In the body of the text, especially in the Results section, it was quite difficult to follow the different mutational hotspots from gene site to gene site, it would have been better to display the most common types in tabular format for quicker views and understanding of the cardinal gene alterations.

Response: Table 3: Mutational status of patients with CC by genes.

Gene alterations Number (%) 

KRAS (n=210) :

Wildtype (n=133)

Mutated (n=77) 

66.3% 

36.7% 

NRAS (n=210) : 

Wildtype (n=204)

Mutated (n=6) 

97.1% 

2.9% 

BRAF (n=200) : 

Wildtype

Mutated 

100% 

 0% 

-The impetus to investigate the WT and mutated RAS gene sites was that mutations in KRAS outside exon 2 (KRAS exons 3,4) and in other downstream effectors of the EGFR signaling pathway (like NRAS) were published to be associated with resistance to anti-EGFR monoclonal antibodies that were used as a standard regimen for CRC patients – meaning that the two Anti-EGFR monoclonal antibodies, panitumumab and cetuximab, used to treat CRC patients with KRAS and NRAS mutations, were not effective. Therefore, the policy was to successfully treat CRC patients, the full RAS WT status had to be confirmed in all mCRC patients before initiating treatment. The question is: Does this study conclusively resolve that issue? If so, that must be clearly stated in the manuscript.

Response: this point has been reported in methods section; Since 2016, we have started testing complete RAS status systematically, for every metastasis colon cancer patient when considered for anti-EGFR therapy.

-Compared to NRAS wild-type patients, “NRAS-mutated patients seem to be exhibiting lower total lymph node rate, and were associated with higher lymph node metastasis number”. This statement is confusing. It could be revised to make the right statement without seemingly contradicting itself. 

Response: Compared to NRAS wild- type patients, NRAS-mutated patients, were more likely to exhibit lower total lymph node rate (P=0.04). This subgroup of tumors was also marked by the presence of positive lymph node (83.3%, P=0.01).

-Among RAS mutations, KRAS codon 13 mutations seemed to have a poor prognostic value, which may negatively impinge on overall survival of all the stages of CRC patients. Is that a correct statement?

Response: this statement has been reviewed; KRAS codon 13-mutated status was the worst predictor of prognosis at all stages in our population.

---

## [Decision Letter · Decision Letter 1]

1 Mar 2021

Mutation status and prognostic value of KRAS and NRAS mutations in Moroccan colon cancer patients: a first report.

PONE-D-20-28747R1

Dear Dr. FATIMA EL AGY

We’re pleased to inform you that your manuscript has been judged scientifically suitable for publication and will be formally accepted for publication once it meets all outstanding technical requirements.

Kind regards,

Hassan Ashktorab

Academic Editor

PLOS ONE

Additional Editor Comments (optional):

Reviewers' comments:

Reviewer's Responses to Questions

**Comments to the Author**

1. If the authors have adequately addressed your comments raised in a previous round of review and you feel that this manuscript is now acceptable for publication, you may indicate that here to bypass the “Comments to the Author” section, enter your conflict of interest statement in the “Confidential to Editor” section, and submit your "Accept" recommendation.

Reviewer #2: All comments have been addressed

2. Is the manuscript technically sound, and do the data support the conclusions?

Reviewer #2: Yes

3. Has the statistical analysis been performed appropriately and rigorously? 

Reviewer #2: Yes

4. Have the authors made all data underlying the findings in their manuscript fully available?

Reviewer #2: Yes

5. Is the manuscript presented in an intelligible fashion and written in standard English?

Reviewer #2: No

6. Review Comments to the Author

Reviewer #2: This is a much improved manuscript. However, the English grammar issue still persist throughout the manuscript and especially in the Introduction section stating with the first sentence where "in" before the word "transduce" must be taken out. Also, somewhere further down the paragraphs "gander" was used for "gender..

7. PLOS authors have the option to publish the peer review history of their article (what does this mean?). If published, this will include your full peer review and any attached files.

Reviewer #2: No

---

## [Editor Report · Acceptance letter]

9 Mar 2021

PONE-D-20-28747R1 

Mutation status and prognostic value of KRAS and NRAS mutations in Moroccan colon cancer patients: a first report 

Dear Dr. El agy:

I'm pleased to inform you that your manuscript has been deemed suitable for publication in PLOS ONE. Congratulations! Your manuscript is now with our production department. 

Kind regards, 

on behalf of

Dr. Hassan Ashktorab 

Academic Editor

PLOS ONE